# Area-Level Determinants in Colorectal Cancer Spatial Clustering Studies: A Systematic Review

**DOI:** 10.3390/ijerph181910486

**Published:** 2021-10-06

**Authors:** Sharifah Saffinas Syed Soffian, Azmawati Mohammed Nawi, Rozita Hod, Huan-Keat Chan, Muhammad Radzi Abu Hassan

**Affiliations:** 1Department of Community Health, Faculty of Medicine, Universiti Kebangsaan Malaysia, Kuala Lumpur 56000, Malaysia; p108896@siswa.ukm.edu.my (S.S.S.S.); rozita.hod@ppukm.ukm.edu.my (R.H.); 2Clinical Research Center, Sultanah Bahiyah Hospital, Alor Setar 05400, Malaysia; huankeat123@yahoo.com (H.-K.C.); drradzi91@yahoo.co.uk (M.R.A.H.)

**Keywords:** colorectal cancer, risk factors, cluster, geographical information system, ecobiosocial

## Abstract

The increasing pattern of colorectal cancer (CRC) in specific geographic region, compounded by interaction of multifactorial determinants, showed the tendency to cluster. The review aimed to identify and synthesize available evidence on clustering patterns of CRC incidence, specifically related to the associated determinants. Articles were systematically searched from four databases, Scopus, Web of Science, PubMed, and EBSCOHost. The approach for identification of the final articles follows PRISMA guidelines. Selected full-text articles were published between 2016 and 2021 of English language and spatial studies focusing on CRC cluster identification. Articles of systematic reviews, conference proceedings, book chapters, and reports were excluded. Of the final 12 articles, data on the spatial statistics used and associated factors were extracted. Identified factors linked with CRC cluster were further classified into ecology (health care accessibility, urbanicity, dirty streets, tree coverage), biology (age, sex, ethnicity, overweight and obesity, daily consumption of milk and fruit), and social determinants (median income level, smoking status, health cost, employment status, housing violations, and domestic violence). Future spatial studies that incorporate physical environment related to CRC cluster and the potential interaction between the ecology, biology and social determinants are warranted to provide more insights to the complex mechanism of CRC cluster pattern.

## 1. Introduction

Cancer is one of the most important causes of mortality and morbidity around the globe. It is the third cause of death after cardiovascular diseases and road traffic injuries as reported by the Global Burden of Disease Study 2017 [1]. Among the types of cancers, colorectal cancer (CRC) accounted as the third most common cancer in men and the second highest in women [2]. The disease represented the loss of 15,800,000 disability-adjusted life years in 2013, 56% of which in middle- and low-income countries and 44% in the industrialized countries [3]. While the trend of colorectal cancer had shifted to the left in the western regions, the new cases diagnosed among the young and elderly age group Asians are increasing [4,5].

CRC is multifactorial by nature. No single hazardous factor is plausibly related to CRC, but individual factors such as sex, age, and family history, lifestyle behaviors including alcohol consumption, high intake of red meat and processed meat, low fruit and vegetable intake, high-fat diet, and physical inactivity were massively studied [6,7,8]. Few researchers postulated on the specific gene–environment interaction likely to cause CRC, on an individual basis [9,10]. Although education on the risk factors for CRC have been continuously delivered to the public, there are still evidences of huge disparities in CRC incidence across different location descriptively [11,12].

While many factors were found related to CRC, generally they can be classified into modifiable and non-modifiable. Among the established modifiable risk factors include obesity, westernized diet, physical inactivity, and low fiber intake [7,13,14,15]. Meanwhile, the non-modifiable risk factors include hereditary, age, gender, and ethnicity [16,17,18]. Numerous studies in the literature have focused on colorectal cancer because of its high incidence and mortality and that it is closely related to individual lifestyle (modifiable risk factors), indirectly the tendency to cluster. People living in the same neighborhood tend to have similar lifestyle and share many cluster-inducing factors that grow substantial public concern over locally elevated CRC incidence [19,20]. Despite of the knowledge on modifiable and non-modifiable risk factors, less is known about interaction between multiple risk factors that may occur within a small geographical area. Thus, there has been an expansion of studies that explore the relationship between simultaneous risk factors collectively contributing to the potential of CRC cluster within a local area.

According to the classical theory of epidemiological triad, disease transmission can be explained through host, agent, and environment factors and the interaction that overlapped with each other. The triad model had shown successful intervention to curb the spread of infectious disease. The recognition of multiple risk factors related to CRC proposes the implementation of ecobiosocial approach to explain the relationship and interaction between the environment, biological, and social. Historically, the concept of ecobiosocial was essential in the field of vector-borne diseases, whereby integrated vector management actions and planning were first developed [21]. In 2016, an obesity framework based on the ecobiosocial concept was attempted to further elucidate the significance of environmental influence that shape the unhealthy choice of foods, compounded with social and genetic factors leading to the issue of obesity [22]. As CRC is a chronic disease, requiring more than a decade of potential exposure to the multiple risk factors, it may result from the complex interaction of ecobiosocial factors. The ecological was referred to as “individual activity” and “activity environment” [22] that disrupt the energy imbalance promoting obesity, one of the CRC risk factors. This includes any factors that promote or inhibit the physical activity of an individual within a local setting such as the availability of recreational park, green areas, safe walkability, or accessible public transportation as provided in many major cities. The biological component features the predisposing genetic factors, age, sex, ethnicity, and obesity depicted the susceptibility of an individual towards developing CRC. Notwithstanding that, the social factors define the behaviors and attitudes are likely shaped by the cultural importance of eating pattern, westernized diet, socioeconomic status, smoking habit, and lack of health-seeking behavior.

Spatial cluster detection is an important tool in cancer surveillance to identify areas of elevated risk and to generate subsequent hypotheses about cancer etiology [23,24]. Provided that the existence of known established risk factors of an area, spatial cluster analysis may predict the future trend of cancer locally and inform control strategies. A spatial disease cluster is defined as an area with an unusually higher disease incidence rate [25]. However, the term has been vaguely used to refer to a population-based cancer epidemiology due to the complex interaction between multiple factors believed to contribute to such event. Cancer cluster identification is heavily dependent on the accuracy of methodological design used to estimate the local relative risk as compared to the control [26,27]. Besides, spatial analysis of CRC incidence may provide a new knowledge on the relationships between external risk factors and people lifestyle with CRC burden across communities. This will enable policymakers to develop tailored intervention to areas where the CRC risk is greater. Therefore, the review aimed to identify and synthesize available evidence on clustering patterns of CRC incidence, specifically related to the associated determinants.

## 2. Materials and Methods

The systematic review was conducted in compliant to the PRISMA or Preferred Reporting Items for Systematic Reviews and Meta-Analyses review protocol [28]. PRISMA aims to guide researchers source the appropriate information at high level of accuracy. Based on the protocol, the authors initiated the systematic literature review by formulating appropriate research questions. The authors performed systematic searching that consists of identification, screening, and eligibility. The authors then proceed to appraise the quality of the selected articles using the Appraisal tool for Cross-Sectional Studies (AXIS tool) [29] to ensure the quality of the articles included. Upon completion, the authors read through in detail all the articles for data extraction and analysis.

### 2.1. Formulation of the Research Question

The research question was formulated based on the PICO concept; a tool often used to assist authors in developing suitable research questions for the review. It consists of Population or Problem, Interest, and Context/Outcome [30]. Based on this concept, the authors have included the three main aspects in the review: colorectal cancer (Population), spatial cluster (Interest), and determinants (Context/Outcome), which led the authors to the main research question “What are the determinants commonly linked to colorectal cancer cluster in spatial studies?”.

### 2.2. Systematic Searching Strategies

The systematic searching strategy preceded by the identification, screening, and eligibility stages (Figure 1).

### 2.3. Identification 

Identification stage involved enrichment of the keywords through the utilization of synonyms and their variation to be used during article searching in the databases. The search string was developed and computed using Boolean operators and phrase searching as shown in Table 1. The systematic literature search was conducted between 24 and 27 May 2021, which involved four primary databases: Scopus, Web of Science, PubMed, and EBSCOHost, resulted in the retrieval of 3134 records. These four databases were selected because of their availability and accessibility to our institution. There were 45 duplicate records found and removed. The records were exported from the databases and arranged for screening in an Excel sheet.

### 2.4. Screening 

The title and abstract of each article were examined for relevance and screened based on specific criteria by the authors. The inclusion criteria for article selection were: (1) published between 2016 and 2021, (2) full original article, (3) written in English, (4) study focused at identifying colorectal cancer clusters or interrelations between two or more of the clusters. The duration of published articles screened was determined based on the recent development and dynamicity of Geographic Information System (GIS) software. Articles of systematic review, conference proceeding, book chapter, and reports were excluded. Any disagreement on article selection was resolved via discussion. The screening process had excluded 1603 articles, while the remaining 46 articles proceeded for retrieval of full text for eligibility.

### 2.5. Eligibility 

There were 38 full text articles successfully retrieved for eligibility. The authors reviewed all full text articles and recorded the reason for the article exclusion. A total of 26 articles were excluded due to the absence of spatial analysis (n = 13), the articles focus on CRC mortality (n = 5), focus on CRC screening adherence (n = 5) and article that combined other type of cancer (n = 3). The remaining articles were resumed with the quality appraisal process.

### 2.6. Quality Appraisal 

The articles selected from the eligibility process must be further examined for risk of bias assessment to ensure the quality of the study [31]. Study quality was assessed using the appraisal tool for observational and cross-sectional studies (AXIS tool) as shown in Table 1. The scale is designed for non-experimental research and includes 20 items measuring each aspect of study quality [29]. Each study was assessed for potential risk of biases through key domain areas of study design, sample size justification, target population, sampling frame, sample selection, measurement validity and reliability, methodology limitations and discussion. Two authors conducted the assessments independently. Any disagreement between the two authors was resolved through discussion until consensus met; when necessary, a third reviewer was consulted. A total of 12 articles were included in the final stage.

The result for quality assessment is presented in Table 2. The total number of “yes” were recorded for every study as the tool guide does not specify the standardize scoring measure. The mean total quality score was 15.4 (range 14–16). Of the 12 studies finalized, quality assessment using AXIS revealed all the included studies had clear study objectives and employed appropriate study design with respect to their objectives. Similarly, all 12 studies clearly defined the target population with an appropriate sampling frame. Only three studies addressed and categorized non responders [32,33,34]. The risk factor and outcome variables measured were appropriate to the aims of each study and were correctly measured. All studies clearly explained the statistical significance used and sufficiently described the methodology to enable them to be repeated. One study [35] inadequately described the basic data. Meanwhile, none of the studies reported information on the nonresponders, possibly due to the nature of ecological analysis used in spatial studies. All studies provided information on the methodological limitations. Five studies did not state any information regarding ethical approval or consent of participants [33,36,37,38,39].

### 2.7. Data Abstraction and Analysis

The authors independently extracted information from the included studies on author’s names, year, country, study objectives, study designs, sample, statistical test used, factors associated with cluster, additional findings, and limitations. Discrepancies were resolved through discussion between the two authors. Following that, the authors systematically synthesize the outcomes based on the use of words and text to explain the findings according to patterns, themes, consistencies, inconsistencies, and relationship within the extracted data [40].

Data abstracted from all studies were recorded in an appropriate matrix table (Table 3). The authors reviewed the matrix tables for consistencies and inconsistencies to generate themes and findings for the review. Similar or related information were grouped together as one characteristic and the technique repeated to form reasonable findings for interpretation. The authors discussed respective thoughts associated with findings until the point of agreement was reached upon adjustment of the generated findings.

## 3. Results

A total of 12 studies were included in this systematic review (Table 3). Descriptive summary of included studies concerning study location, setting and study design are shown in Figure 2. All eligible studies were conducted spanning five countries that include Canada [41], United States [11,32,42], France [43], Portugal [35], and Iran [33,34,36,37,38,39]. Comparing study locations based on the WHO regions, four studies [11,32,41,42] accounted for the Region of the Americas (AMR), two studies [35,43] from the European Region (EUR) and six studies [33,34,36,37,38,39] were performed in the Eastern Mediterranean Region (EMR). The analyzed articles were published in the year 2016 to 2021. Seven of these quantitative studies resumed cross sectional study design [33,34,35,39,41,42,43], two studies were conducted using retrospective cohort [11,32], and three articles of ecological studies [36,37,38]. 

The period of data collection for a quarter of the included studies dated back before 2000 [32,41,43] whereas nine more ranges between 2000 to 2017 [11,33,34,35,36,37,38,39,42]. The duration of retrospective data collection can be categorized into less than 10 years [11,33,34,35,36,37,38,39,42,43], more than 10 years [32] and more than 20 years [41]. Spatial analysis was applied to address a range of objectives (Table 3), with the most common include description of the CRC distribution, analysis of associated factors [11,32,33,36,38,39,41,42], and comparison of different cluster detection method [43].

### 3.1. Type of Spatial Analysis

The review showed multiple types of spatial cluster statistics being used across the included studies. Seven studies [11,34,35,38,39,42,43] utilized Moran’s Index to summarize the spatial autocorrelation over study area, three studies [32,36,41] used Poisson Regression Model, three studies [11,32,37] used Getis–Ord Gi, two studies [35,42] used local indicators of spatial association (LISA) and each one study analyzed their data using at least Besag–York–Mollie (BYM) model [38] and Generalized Linear Models [33], respectively or in combination with the others. Generally, the type of test can be classified into global, local, and focused tests according to the study hypotheses. The global cluster statistics, such as Moran’s I often used to inform the existence of spatial structure of an area, not considering the point of location or the difference between different cluster [44]. Meanwhile, local statistics such as LISA and Ord–Getis Gi, explained on the nature of the spatial dependency of a given locality and focused test (e.g., Poisson Regression Model) explore the possible clusters near potential risk factors [44].

### 3.2. Factors Associated with CRC Cluster

Evidence of clustering were abundance as most of the studies reported presence of CRC cluster in the study population. However, the outcomes were more meaningful when the studies incorporate other factors to further understand the association with CRC cluster. The review found multiple factors frequently studied such as age, sex, ethnicity, overweight and obesity, smoking, daily consumption of fruit and milk, socioeconomic status as represented by the median income level, employment status, health costs, housing violations or domestic violence, health care coverage, urbanicity, dirty streets and tree coverage. Collectively, these can be summarized into social, biology and ecology determinants (Table 4). 

When comparing the factors associated with CRC cluster in all the included studies, eight studies [11,32,33,36,38,41,42,43] defined the relationship of CRC cluster with social factors, another nine studies [32,34,35,36,37,38,39,41,42] explain on the biology factors, while two studies [11,42] analyzed on the ecological factor. 

While CRC has been commonly linked to westernized diet and physical inactivity, less focus was given to explore on the ecological factor leveraging towards colorectal carcinogenesis. Two studies suggest that the surrounding physical environment has temporally shaped the progression of CRC cluster within an area [11,42]. High accessibility to healthcare facilities was found correlated with substantial CRC screening rate, hence increase in CRC incidence and cluster [42]. Similarly, the CRC clusters were found dependent with the urbanicity level of an area [42]. The definition of an urban area by Kuo et al. (2019) was determined based on the population density of each county, which can be misleading when applied to contextual geographic variation due to ecological factor [42]. Fast food outlets offering high-dense fat diet and processed meat were more common in urban cities, thus worth explore to predict future CRC cluster. Even though factors like dirty streets and tree coverage as proxy to the physical environments were incorporated into the scoring of the community statistical area characteristics, the role of ecology per se was not highlighted [11]. Therefore, future studies to explore on the influence of ecological factors and CRC cluster is recommended.

Eight studies [11,32,33,36,38,41,42,43] applied the spatial autocorrelation approach to explore the influence of social factors on CRC cluster and found out that neighborhoods with higher median household income level ranging between USD 38,040 and USD 80, 876 annually in 2011, were associated with decreased risk of both early and late-stage CRC [32]. On the other hand, in the middle-income countries where universal health coverage is an issue, the high CRC cluster pattern was observed with more frequent utilization of health services as measured through the health cost. Indirectly, the association explained the impact of socioeconomic inequalities against CRC incidence over time [36,43]. However, further information regarding the tumor stage following early detection and treatment deem important is lacking to complement the circumstances. Besides that, the aggregation analysis makes it difficult to elicit causal effect linkage on individual basis with regards to respective economic background.

The biological factors frequently analyzed in CRC spatial cluster studies were age and sex. Geographical areas with ageing population tend to form CRC cluster as compared to the younger age group [34,39,41]. However, information on the length of residency and migration activities were lacking to verify the plausible relationship between age and CRC cluster in the context of residential areas [11,42]. On the other hand, Pakzad et al. (2016) and Roquette et al. (2019) revealed specific spatial pattern of CRC cluster for men and women respectively [35,37]. The findings suggest for potential sex-specific determinants susceptible to CRC in certain areas despite the exposure to several other risk factors such as high fat diet, physical inactivity, and smoking. In areas with heterogenous ethnic population, Liu et al. (2016) reported higher CRC incidence among the Hispanics than non-Hispanics Whites and Blacks [32]. The selective trend against particular ethnicities likely supports the notion of gene–environment interaction in the progression of CRC that may arise from culturally specific dietary pattern and lifestyle. However, other factors such as the length of residency and social reciprocity should be critically considered and controlled with the native population. 

Based on the findings, there is compelling evidence for future research on the interaction of ecological, biological, and social factors collectively with the geographic distribution of CRC incidence, to create area-level tailored cancer care services. Through the baseline information on the local patterns of CRC distribution, allocation of resources could be made available and planning of more targeted community intervention.

## 4. Discussion

The review systematically identified frequent reference made to coin the population-based CRC cluster through methodological approach. Differences in methodology and statistical methods used were described to gauge better understanding of spatial cluster definition [11,45]. To date, there is little consensus on the definition of cluster pertaining to non-communicable diseases specifically colorectal cancer in the community [46]. Whereas the CDC defines a cancer cluster as “a greater than expected number of cancer cases that occurs within a group of people in a geographic area over a defined period of time”, there is notoriously vague and grey area on the definitive baseline figure for such cancer in an area [25,47]. The challenges elicited upon investigation of cancer clusters at fields had called for multiple arguments on the validity of the statistical analyses [27,48]. To overcome the potential inflated probability, this has called for alternative approach of using a standardized incidence ratio (SIR) analysis based on causal inference framework rooted for cancer cluster [49]. The statistical constructs suggested for exposure hypotheses as compared to the traditional observed cancer outcomes shed some light to more reliable findings.

Spatial cluster analysis plays an important role in quantifying geographic variation patterns [26,42,50]. It is commonly used in disease surveillance, spatial epidemiology, population genetics, landscape ecology, and many other fields, but the underlying principles are the same [51,52]. Spatial patterns are of interest to be used in cancer research to explain the link between exposure to the surrounding environments and development of cancer over more than ten years, of which the existing environment and ecosystem might have undergone drastic changes. Several approaches to the geographic pattern recognition include visualization techniques based on “eye-balling”, kernel-based methods that accentuate differences on a surface, artificial intelligence approaches and exploratory spatial data analysis (ESDA) which rely on statistical test [44].

Despite the progress in spatial statistics utilized in CRC research, the review highlighted that many studies are complacent with the traditional Moran’s I index to examine the spatial independence. The frequent usage may be related to the universal understanding of similar interpretation behavior relatively when compared to correlation coefficient [53]. With the growing interest of spatial statistics methodologies employed in various contexts, the heterogeneous geospatial studies of CRC incidence have led to difficulty in the comparison of study outcomes [32,37,38]. Spatial analysis was applied to improve understanding of a range CRC-related issues, including the distribution and determinants, the mechanisms driving the local CRC epidemiology, the effect of preventive strategies and the barriers to seek for treatment. Often, the geospatial methods have been combined with environmental factors exposure to understand the drivers of local cancer epidemiology; however, such studies remain limited for CRC in high-incidence areas [11,42,54,55]. 

Whereas the ecological determinants showed great significance when applied in spatial research, this factor has been lacked studied in relation to CRC. Factors such as health care coverage, urbanicity, dirty streets, and tree coverage were analyzed as the ecological determinants in the review. High accessibility of healthcare facilities offers better services including screening, thus concomitant with high CRC cluster [42,56]. Many studies linked urbanicity with higher CRC incidence partly due to the availability of health infrastructure and advance treatment options, besides the highly dense population. Walkability areas and greenness of streets have been associated with the amount and duration of physical activity [57], one of the well-known protective factors for CRC.

Exposure to unhealthy food environments such as the availability and accessibility to unhealthy food stores may encourage the surrounding community to have less healthy diets [54,58,59]. Likewise, the absence of green spaces or recreational parks nearby for physical activity may lead to continuous physical inactivity [19,60,61]. These are the examples of physical environment potentially instill the unhealthy lifestyle to localized settings, which may pose higher risk towards CRC in long-term. The review identified minimal studies that explore the association of the physical environment with population-based CRC cluster [11,42]. Physical environment plays important role that influence the formation of obesogenic environment, shaping the behavior and lifestyle of the population [62]. It contributes both direct and indirect pathways towards occurrence of CRC. Previous literatures highlighted the significance of ecological factors’ exposure with CRC clusters to justify public health actions and policy in the context of preventive strategies [14,54,60,61,63]. Inability to identify potential modifiable factors within the physical environment poses salient challenges towards future CRC prevention and control.

Previous literatures examined molecular genetics found associated with CRC. This includes the APC gene, K-ras family, p53, DCC, and several mismatched repair genes leading to mutations throughout the genome of affected cells. While more than 50% of sporadic CRC cases were linked to some degree of genetic mutation, the occurrence of COX-2 genetic polymorphism is particularly high among Caucasians compared to the Asians [64]. Males are more likely than females to be diagnosed as CRC across all age groups, demonstrating the role of sex in carcinogenesis [64,65]. 

Other social aspects such as adherence to the existing CRC screening program may provide insight to explain the existence of a local cluster pattern [66,67]. A high-incidence area for CRC can be due to large uptake of CRC screening by the people, indicating good health awareness [68]. The behavior and attitude towards health highly influenced by the socioeconomic status [69]. Many studies supported better health outcome in countries providing universal health coverage [41]. Similarly, few recent spatial studies showed high–high clusters of CRC concentrated in urban areas compared to rural areas [11,34,42]. This can be further explained by the urbanized lifestyle that leads to more readily accessible and available online food delivery at present [70], promoting physical inactivity across all age groups in the population. Thus, future studies that examine the influence of urbanized lifestyle with the formation of CRC spatial cluster is recommended.

The ecological, biological, and social determinants have significant impacts to formation of geographical aggregated pattern of CRC incidence to an extent, when studied independently. In circumstances of the true population, most of the factors present simultaneously and possibly interact with each other, producing greater effect to increased risk of CRC. Although major studies highlighted the synergistic effect at the individual level, through animal studies supporting the gene environment interaction [71,72], there have been limited studies that examine the interaction between these factors to benefit the preventive strategies at the population level. The combined effect analysis is crucial to inform the multisectoral stakeholders on the challenging CRC burden as a shared public health issue. Figure 3 summarizes the interaction occurring between ecological, biological, and social determinants that potentially influence the existence of CRC cluster within population.

### 4.1. Strength

The review identified potential future research area on the association of ecological, biological, and social factors with the clustering pattern of CRC incidence. With the existing knowledge on population CRC cluster driven by various sociodemographic circumstances, future studies can be designed to explore the physical and built environment across various geographic settings. Furthermore, it provides insight to the multilevel stakeholders on more specific intervention and preventive strategies tailored to the high-risk areas.

### 4.2. Limitation

While ecological, biology, and social determinants are interrelated to cause colorectal cancer both at the individual level and community level, it is difficult to distinguish the relations of each factor independently. Spatial analysis studies focusing on cancer incidence secondary to occupation-related were not included in the review, thus limit the discussion on ecological and social influence towards cluster distribution. 

Most of the reviewed articles were from middle-income settings, which may either reflect publication bias or a focus of research efforts on such settings. In high-incidence countries of the Asian region, studies with limited use of spatial analysis methods could reflect a lack of access to information resources or insufficient expertise in these settings. Nonetheless, the review revealed areas with high CRC incidence stand to gain the most from understanding of CRC spatial patterns in which clustering may be important epidemiologically.

Nearly all the models have shown significant associations between CRC cluster and demographic, socioeconomic, and risk-factor variables, although is it difficult to rule out publication bias favoring studies with positive findings. However, associations observed between CRC cluster and different factors such as sex, household income, and obesity at the population level vary across studies. These were recognized as important individual-level risk factors, highlighting the potential for ecological fallacy.

## 5. Conclusions

In conclusion, the review identified robust evidence of CRC cluster across different geographical settings. However, attempts to examine the association of area-level determinants and CRC cluster are lacking in ecology as compared to the common biology and social attributes. Therefore, future spatial studies that incorporate physical environment (ecological) factors in this research field are warranted as guides for policymakers to plan more targeted preventive and control actions. Studies relating more than one determinant with the CRC cluster displayed potential degree of interaction, which is understudied. Future interaction analysis that incorporates the combination of ecology, biology, and social attributes may benefit to explain the trend of CRC cluster in detail, thus validating the cancer control continuum planning.

## Figures and Tables

**Figure 1 ijerph-18-10486-f001:**
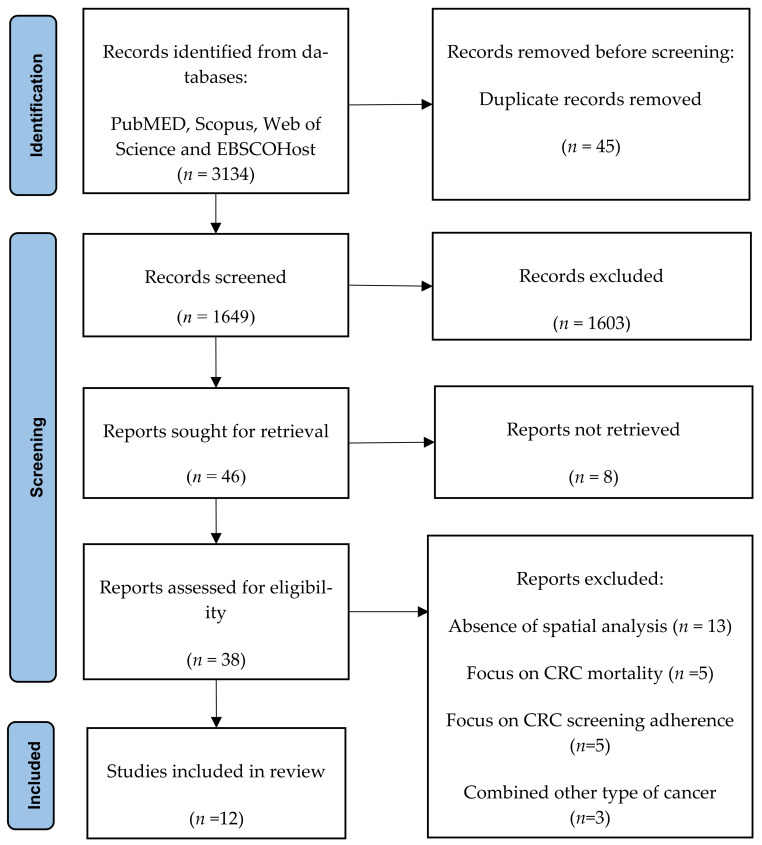
PRISMA flow diagram.

**Figure 2 ijerph-18-10486-f002:**
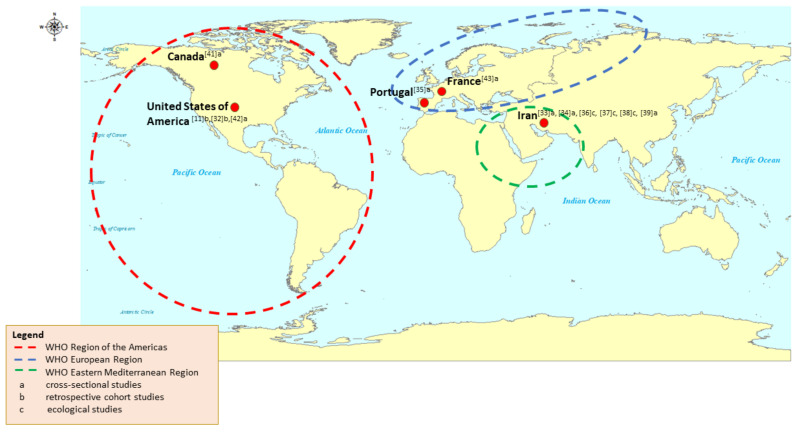
Distribution of included articles based on countries, WHO regions and study design.

**Figure 3 ijerph-18-10486-f003:**
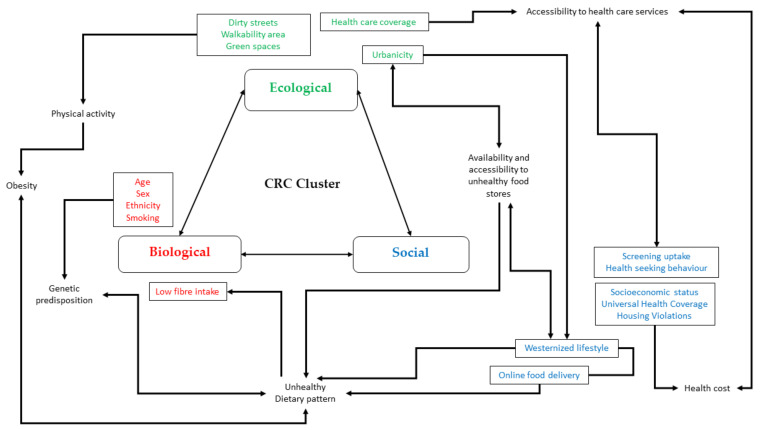
Conceptualization of interaction between ecological, biological, and social determinants influence formation of CRC cluster.

**Table 1 ijerph-18-10486-t001:** Keyword search used in the identification process.

Database	Search String
Scopus	TITLE-ABS-KEY ((“colorectal cancer” OR “colorectal neoplasm * ^1^” OR “colorectal tumor *” OR “colorectal carcinoma” OR “large bowel cancer”) AND (“cluster analysis” OR “spatial analysis” OR “geographical information system” OR “geographic distribution” OR “incidence distribution” OR “demography”)) AND (“risk factor *” OR “cancer risk” OR “determinant *”))
Web of Science	TS = ((“colorectal cancer” OR “colorectal neoplasm *” OR “colorectal tumor *” OR “colorectal carcinoma” OR “large bowel cancer”) AND (“cluster analysis” OR “spatial analysis” OR “geographical information system” OR “geographic distribution” OR “incidence distribution” OR “demography”)) AND (“risk factor *” OR “cancer risk” OR “determinant *”))
PubMed	((“colorectal cancer” OR “colorectal neoplasm *” OR “colorectal tumor *” OR “colorectal carcinoma” OR “large bowel cancer”) AND (“cluster analysis” OR “spatial analysis” OR “geographical information system” OR “geographic distribution” OR “incidence distribution” OR “demography”)) AND (“risk factor *” OR “cancer risk” OR “determinant *”))
EBSCOHost	((“colorectal cancer” OR “colorectal neoplasm *” OR “colorectal tumor *” OR “colorectal carcinoma” OR “large bowel cancer”) AND (“cluster analysis” OR “spatial analysis” OR “geographical information system” OR “geographic distribution” OR “incidence distribution” OR “demography”)) AND (“risk factor *” OR “cancer risk” OR “determinant *”))

^1^ The symbol * was used in the search strategy as truncation and wildcard function to increase variability of selected keywords.

**Table 2 ijerph-18-10486-t002:** Quality assessment of included studies using AXIS tool.

Articles	Introduction	Methods									
Author (Year), Country	Were the aims/objectives of the study clear?	Was the study design appropriate for the stated aim(s)?	Was the sample size justified?	Was the target/reference population clearly defined?	Was the sample frame taken from an appropriate population base so that it closely represented the target/reference population under investigation?	Was the selection process likely to select subjects/participants that were representative of the target/reference population under investigation?	Were measures undertaken to address and categorise non-responders?	Were the risk factor and outcome variables measured appropriate to the aims of the study?	Were the risk factor and outcome variables measured correctly using instruments/measurements that had been trialled, piloted, or published previously?	Is it clear what was used to determine statistical significance and/or precision estimates?	Were the methods (including statistical methods) sufficiently described to enable them to be repeated?
Liu et al. (2016),United States	Y	Y	Y	Y	Y	Y	Y	Y	Y	Y	Y
Mansori et al. (2018), Iran	Y	y	y	y	y	y	CT	Y	Y	Y	Y
Mansori et al. (2019), Iran	Y	Y	Y	Y	Y	Y	CT	Y	Y	Y	Y
Pakzad et al. (2016), Iran	Y	Y	Y	Y	Y	Y	CT	Y	Y	Y	Y
Pourhoseingholi et al. (2020), Iran	Y	Y	Y	Y	Y	Y	Y	Y	Y	Y	Y
Goungounga et al. (2016), France	Y	Y	Y	Y	Y	Y	CT	Y	Y	Y	Y
Roquette et al. (2019), Portugal	Y	Y	Y	Y	Y	Y	CT	Y	Y	Y	Y
Torres et al. (2018), United States	Y	Y	Y	Y	Y	Y	CT	Y	Y	Y	Y
Halimi et al. (2019), Iran	Y	Y	Y	Y	Y	Y	Y	Y	Y	Y	Y
Harminder et al. (2017), Canada	Y	Y	Y	Y	Y	Y	CT	Y	Y	Y	Y
Kuo et al. (2019), United States	Y	Y	CT	Y	Y	Y	CT	Y	Y	Y	Y
Goshayeshi et al. (2019), Iran	Y	Y	Y	Y	Y	Y	CT	Y	Y	Y	Y
**Articles**	**Results**						**Discussion**		**Other**		
	Were the basic data adequately described?	Does the response rate raise concerns about non-response bias?		If appropriate, was information about non-responders described?	Were the results internally consistent?	Were the results for the analyses described in the methods, presented?	Were the authors’ discussions and conclusions justified by the results?	Were the limitations of the study discussed?	Were there any funding sources or conflicts of interest that may affect the authors’ interpretation of the results?	Was ethical approval or consent of participants attained?	Total Recorded “Yes”
Liu et al. (2016), United States	Y	N		N	CT	Y	Y	Y	N	Y	16
Mansori et al. (2018), Iran	Y	CT		CT	CT	Y	Y	Y	N	CT	14
Mansori et al. (2019), Iran	Y	CT		CT	CT	Y	Y	Y	N	CT	15
Pakzad et al. (2016), Iran	Y	N		CT	Y	Y	Y	Y	CT	CT	15
Pourhoseingholi et al. (2020), Iran	Y	N		CT	Y	Y	Y	Y	N	CT	16
Goungounga et al. (2016), France	Y	N		CT	Y	Y	Y	Y	N	Y	16
Roquette et al. (2019), Portugal	N	N		CT	Y	Y	Y	Y	N	Y	15
Torres et al. (2018), United States	Y	N		N	Y	Y	Y	Y	N	Y	16
Halimi et al. (2019), Iran	Y	N		N	CT	Y	Y	Y	N	Y	16
Harminder et al. (2017), Canada	Y	N		N	CT	Y	Y	Y	N	Y	15
Kuo et al. (2019), United States	Y	N		CT	Y	Y	Y	Y	N	Y	15
Goshayeshi et al. (2019), Iran	Y	N		CT	Y	Y	Y	Y	N	CT	16

**Table 3 ijerph-18-10486-t003:** Characteristics of included studies.

No.	Author (Year), Country	Objective	Study Design	Sample	Statistical Test	Findings	Factors Associated	Determinants	Limitation
1	Liu et al. (2016), United States	To determine the association between median household income and the risk of developing colorectal cancer in Texas	Retrospective cohort	155,534 men and women with colorectal cancer in 1995–2011	Getis-Ord G, Poisson regression model (ArcGIS v10.1)	Higher median household income, measured from the third to the highest income quintile ranging between $38,040 and $80,876, was associated with decreased risk of colorectal cancer in Texas. The Hispanics showed higher incidence rate of CRC compared to the Non-Hispanic Whites and Blacks at all time period, with slight decrease trend across higher median household income.	County median household income level (Median household incomes in 254 counties classified into five quintiles) Ethnicity (Non-Hispanic Whites, Non-Hispanic Blacks, Hispanics)	Social Biology	Income variable was the median household income measured at county level, thus potential ecological fallacy. Median household income alone may not be a good proxy for SES. Lack of accuracy in population estimates may led to biased calculation for CRC incidence
2	Mansori et al. (2018), Iran	To determine the factors associated with the spatial distribution of the CRC incidence in the neighborhoods of Tehran, Iran	Ecological	2815 new cases of CRC from 2008 to 2011	The Moran Index, BYM model (using OpenBUGS version 3.2.3, ArcGIS 10.3, GeoDa)	There was spatial autocorrelation at the level of the neighborhoods. Significant association was observed between women head of household, living in a rental house, no daily milk consumption in the household and higher household health expenditures against higher risk of CRC respectively.	Socioeconomic variables (SES): unemployed people aged 15 years and above, educated women (university level of education) aged 17 years old and above, women head of household, households without a car, those living in a rental house, households with income below the poverty line, people without insurance coverage. Risk factors: Households without daily fruit consumption, households without daily milk consumption, overweight people aged 15 years and above, smoking households. Health costs: Household health expenditures, expenditures on diagnoses, expenditures on medicine, expenditures on hospitals, expenditures on medical visits	Social Biology	Ecological fallacy. Edge effect, referring to border neighborhoods affected by size of adjacent regions. Misclassification in geocoding due to incomplete postal addresses.
3	Mansori et al. (2019), Iran	To determine effective factors on geographic distribution of the CRC incidence in Iran	Ecological	2815 new cases of CRC from 2008 to 2011	Geographically Weighted Poisson Regression Model (using GWR 4, Stata 14, ArcGIS 10.3)	The spatial variability was observed with more frequent utilization of health services as indicated by the household health expenditures (Median Incidence Risk Ratios (IRR): 1.39), the cost of diagnosis of the disease (Median IRR: 1.03), the cost of household medicine (Median IRR: 1.05), the cost of hospital admission (Median IRR: 1.09) and the cost of medical visits (Median IRR: 1.27)	Socioeconomic variables (SES) include employment status, education Daily fruit consumption, daily milk consumption, overweight, smoking Health costs variables	Social Biology	Ecological fallacy. Unclear addresses for geocoding. Covariates of age, older than 50 years and overweight
4	Pakzad et al. (2016), Iran	To investigate the spatial distribution of colorectal cancer in Iran	Ecological	6210 cases of CRC in 2009	Getis-Ord-Gi* (spatial statistics)	Higher incidence of CRC among men (11.31 per thousand people) than women (10.89 per thousand people) in the northern and central provinces of Iran.	Sex	Biology	Incomplete data registration. Lack of full report Poor data classification
5	Pourhoseingholi et al. (2020), Iran	To determine the distribution of CRC risk in a map with socioeconomic risk factors adjustment	Cross-sectional	21,543 CRC cases between 2005 and 2008	Generalized Linear Model (using WINBUGS program, ArcGIS v10)	Hotspot areas of CRC cluster identified for men were in the North and Western regions (mean SIR 1.92) while the Central provinces reported higher rate for women (mean SIR 1.85). Unemployment rate and mean household income had minimal impacts on CRC cluster.	Unemployment rate (mean ± SD: 11.64% ± 3.18%), Mean household income (mean ± SD: 66.46 Rials ± 12.04 Rials)	Social	Incomplete and lack of up-to-date data on population.
6	Goungounga et al. (2016), France	To compare empirically different cluster detection methods to find spatial clusters of cancer cases	Cross-sectional	3084 CRC cases between 1998 and 2007	Moran’s I, the empirical Bayes index (EBI) and Potthoff-Whittinghill test (using SpODT, SaTScan an HBSM)	The socioeconomic inequalities did not affect the spatial variations of CRC incidence	Socioeconomic disadvantage as proxy by the calculation of Townsend index of deprivation. The index considers the proportion of unemployed people in the workforce, proportion of households without car, proportion of households renting and the proportion of overcrowded households. Increase in the Townsend index indicates an increase in the deprivation level of the inhabitants.	Social	Power of spatial cluster detection methods increases with the event rate
7	Roquette et al. (2019), Portugal	To describe and discuss the geographical patterns of CRC incidence and mortality in Portugal municipalities	cross sectional	37,543 CRC cases between 2007 and 2011	Global Moran’s Index and Local Moran’s Index (LISA), geographically weighted regression (GWR) (using ArcGIS)	The CRC incidence was relatively higher in the Norte region to women than men. Meanwhile, areas in the coastal Centro, the LVT and the Alentejo region showed markedly higher CRC cases among men than to women.	Sex	Biology	Limited data availability on other risk factors
8	Torres et al. (2018), United States	To evaluate the geographic distributions of colorectal cancer incidence among female residents in Baltimore City, Maryland and the neighborhood characteristics associated with those distributions	retrospective cohort	1120 female CRC patients between 2000 and 2010	Spatial clusters identified using Getis-Ord-Gi* statistic and local Moran’s I, global ordinary least square regression model (Using STATA, ArcGIS)	Cluster spot for CRC was identified in two out of 55 Community Statistical Area (CSA) studied. The findings noted that every one percent increase in African-American residents resulted in CRC incidence increasing by 0.031 times per 1000 female residents. The CRC cluster spot experienced less crime with majority residents between the ages of 50 and 74 years old.	The 2012 Baltimore Neighborhood Indicators Alliance Vital Signs report was referred for the neighborhood characteristics. The indicators include females aged 50 to 74 years, percentage of African-American, female-headed households, percentage of households earning less than $25,000, percentage of vacant residential properties, housing violations, number of crime incidents per 1000 residents, count of emergency call for domestic violence, teen birth, employment rates, dirty streets, tree coverage and neighborhood associations.	Social Ecology	Social determinants limited by the residential environment. No information on the length of residency on their addresses.
9	Halimi et al. (2019), Iran	To explore the spatial pattern of CRC incidence in Hamadan province, Iran	Cross-sectional	805 CRC cases during 2007 to 2014	Local Moran’s I (MS Excel, Arc GIS 10.5)	High-high clusters of CRC incidence identified in Mohajeran and Lalejin areas. Majority of the CRC incidence were among men (54%) and those in the age group between 65 and 85 years.	Sex, Age	Biology	Lack of accuracy for registries. Census population data is used
10	Singh et al. (2017), Canada	To determine the variation of CRC incidence by average household income in area of residence	Cross-sectional	19,484 CRC cases between 1985 and 2012	Bayesian Poisson regression models (Using WinBUGS software, ArcGIS v10.3)	There were few small geographic areas in the southwest rural Manitoba with persistent high CRC incidence	Sex, Age, Mean annual household income, proportion of recent immigrants (since 1961 to 2001), rate of visible minority status and unemployment status.	Biology Social	Ecological analysis, hence result should be interpreted in the context of area of residence
11	Kuo et al. (2019), United States	To address spatial autocorrelation between CRC and county-level determinants	Cross sectional	2003 to 2013	Local Indicators of Spatial Association (LISA), Moran’s I (Using ArcMap, SAS, GeoDaSpace software)	The location of high-high clusters identified in the north-eastern counties	Age, Adults with BMI ≥ 30 kg/m^2^ (Obesity), Current smoker adults who smoked at least 100 cigarettes in lifetime, Socioeconomic Status deprivation composed of education level, employment rate, income level, family and social support. Ethnicity studied include non-Hispanic Black, Hispanic, Native American and Asian. Percentage of population aged less than 65 years without health insurance. Urbanicity classified into urban, large town and rural counties.	Biology Social Ecology	Result may not generalize to other geographic areas. Lack of person-level and tumor level data
12	Goshayeshi et al. (2019), Iran	To identify potential spatial factors contributing to its geographical distribution	Cross sectional	1089 CRC cases between 2016 and 2017	Local Moran’s I, Ordinary Least Square Regression (Using MS Excel, ArcGIS v10.6)	CRC clusters identified in Rezashahr, Sarafrazan and Nofel-Loshato areas. The neighborhood of CRC hotspots areas was associated with high proportion of population with 50 years and above, obesity (Body Mass Index ≥30 kg/m^2^), daily fibre intake (≤25 g).	Age, Body Mass Index (BMI), daily consumption of red meat (gram), daily consumption of fibre (gram)	Biology	Study did not consider processed meats. Patients who had shifted were not included. Inability to geocode Persian addresses affect the accuracy

**Table 4 ijerph-18-10486-t004:** Factors analyzed in each of the included studies.

CRC Determinants	Studies
Ecology	
Health care coverage	Kuo et al. 2019
Urbanicity	Kuo et al. 2019
Dirty streets	Torres et al. 2018
Tree coverage	Torres et al. 2018
Biology	
Age	Roquette et al. 2019; Halimi et al. 2019; Singh et al. 2017; Kuo et al. 2019; Goshayeshi et al. 2019
Sex	Pakzad et al. 2016; Roquette et al. 2019; Halimi et al. 2019; Singh et al. 2017
Ethnicity	Liu et al. 2016; Kuo et al. 2019
Overweight and Obesity	Mansori et al. 2018; Mansori et al. 2019; Kuo et al. 2019; Goshayeshi et al. 2019
Daily consumption of fruit	Mansori et al. 2018; Goshayeshi et al. 2019
Daily consumption of milk	Mansori et al. 2018; Mansori et al. 2019
Social	
Smoking	Mansori et al. 2018; Mansori et al. 2019; Kuo et al. 2019
Median household income level	Liu et al. 2016; Pourhoseingholi et al. 2020; Goungounga et al. 2016; Torres et al. 2018; Singh et al. 2017
Health costs	Mansori et al. 2018; Mansori et al. 2019
Employment status	Mansori et al. 2019; Pourhoseingholi et al. 2020; Goungounga et al. 2016; Torres et al. 2018; Kuo et al. 2019
Housing violations, domestic violence	Torres et al. 2018

## Data Availability

Not applicable.

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
