# Peer review of "Area-Level Determinants in Colorectal Cancer Spatial Clustering Studies: A Systematic Review"

_ijerph, 2021, doi:10.3390/ijerph181910486_

Round 1

Reviewer 1 Report

The manuscript review by Sharifah Saffinas Syed Soffian identified robust evidence of CRC clusters across different geographic areas. Authors identify the lack of research studies about the association of area-level determinants and CRC cluster in ecology compared to biology and social attributes as well as the lack of analysis which incorporate the combination of the three (ecology, biology and social) attributes. These studies will be required for the policymakers to develop accurate plans for preventive and control actions. I think that the manuscript is an exhaustive review of the CRC cluster studies and should be published in IJERPH. However, I suggest a few minor changes:

  • Maybe table 4 should be restructured to a visual map. I think that it will be just more visual attractive.
  • Figure 2 is very few informative. I think it should be eliminated and the data just mentioned in the text.
  • Discussion is quite hard to read. I guess that a graphic draw, flowchart or something in that line, could be useful to follow the lecture.

Author Response

Reviewer Comments

Author Response

Maybe table 4 should be restructured to a visual map. I think that it will be just more visual attractive.

The author had converted Table 4 into a visual map indicating the study location and setting, as well as respective study design.

Figure 2 is very few informative. I think it should be eliminated and the data just mentioned in the text.

The author had removed Figure 2. The data was mentioned in the text.

Discussion is quite hard to read. I guess that a graphic draw, flowchart or something in that line, could be useful to follow the lecture.

Figure 3 on the conceptualization of ecological, biological and social determinants influence on CRC cluster was inserted to summarize the discussion.

Reviewer 2 Report

check spelling

check bibliography

Author Response

Reviewer Comments

Author Response

check spelling spelling checked
check bibliography bibliography checked

Reviewer 3 Report

Authors of the manuscript conducted a review about spatial clustering in colorectal cancer. While the introduction, methodology and discussion is well detailed, the presentation of selected articles is very poor.

In the present form the manuscript is not suitable for publishing, heavy re-editing is needed including detailed descriptions of what was found in the selected articles. A basic listing of the selected variables is not sufficient.

Author Response

Reviewer Comments

Author Response

In the present form the manuscript is not suitable for publishing, heavy re-editing is needed including detailed descriptions of what was found in the selected articles. A basic listing of the selected variables is not sufficient.

The author had included further description of findings from the selected articles as presented in Table 3.

Round 2

Reviewer 3 Report

The revised version of the article still has the same issue. Details of the selected articles are only listed in a table. The purpose of a review article is to present the selected topic in a complex way, including all previous findings, limitations through the comparsion of the selected articles.

The main issue with the present manuscript is that while authors listed all the important previous findings, limitations, etc., however, besides the used statistics and the selected factors in those articles, the comparision and "critical organization of previous papers" part, what is an impartant feature of a review article is still missing. The data presented in Table 3 should be re-edited into paragraphs.

Author Response

Text explaining the comparison between articles and critical organization of previous papers was inserted in line 260 to 306.